# Root Morphology and Biomass Allocation of 50 Annual Ephemeral Species in Relation to Two Soil Condition

**DOI:** 10.3390/plants11192495

**Published:** 2022-09-23

**Authors:** Taotao Wang, Lei Huang, Xuan Zhang, Mao Wang, Dunyan Tan

**Affiliations:** College of Life Sciences, Xinjiang Agricultural University, Urumqi 830052, China

**Keywords:** root morphological traits, root economics spectrum, biomass allocation, annual ephemerals

## Abstract

Different organ morphologies determine the manner in which plants acquire resources, and the proportion of biomass of each organ is a critical driving force for organs to function in the future. Regrettably, we still lack a comprehensive understanding of root traits and seedling biomass allocation. Accordingly, we investigated and collected the seedling root morphological traits and biomass allocation of 50 annual ephemeral species to clarify the adaptation to environment. The findings of this study showed that there was a significantly negative correlation between root tissue density (RTD) and root diameter (RD) (*p* < 0.05), which did not conform to the hypothesis of the one-dimensional root economics spectrum (RES). On this basis, we divided 50 plant species into those rooted in dense or gravelly sand (DGS) or loose sand (LS) groups according to two soil conditions to determine the differences in root strategy and plant strategy between the two groups of plants. Our study revealed that the species rooting DGS tend to adopt a high penetration root strategy. However, the species rooting LS adopt high resource acquisition efficiency. At the whole-plant level, 50 species of ephemerals were distributed along the resource acquisition and conservation axis. Species rooting DGS tend to adopt the conservation strategy of high stem biomass fraction and low resource acquisition efficiency, while species rooting LS tend to adopt the acquisition strategy of high root and leaf biomass fraction and high resource acquisition efficiency. The research results provide a theoretical basis for the restoration and protection of vegetation in desert areas.

## 1. Introduction

As a vegetative organ in direct contact with the soil, the capacity of the root system to capture water and nutrients in the soil considerably affects the biomass accumulation and morphology of the above-ground part of the plant, which ultimately impacts the survival and development of the whole plant [1,2]. Furthermore, the root system plays a variety of pivotal functions including physical anchoring, storage, and transportation [3]. Therefore, the root system is an important underground part of plants and plays an essential role in biogeochemical cycling and stabilizing organic matter in soils [4]. The effects of roots on plant and ecological processes are the result of changes in root morphological, chemical, and physiological traits [5].

The resource economics hypothesis (REH) indicates that high specific root length (SRL) in a species indicates resource acquisition, while high tissue density and thick roots indicate high resource conservation [6,7]. The hypothesis assumes that root traits are analogous to leaf traits and therefore coordinate along the one-dimensional root economics spectrum (RES), which represents the trade-off between the acquisition and conservation traits [8,9]. However, some studies have suggested that the relationship between root traits is confounding, which does not concur with the hypothesis of RES [10,11]. Associated with the complexity of the root distribution environment and the diversity of exerting functions, the variation of root traits may be multidimensional [12,13,14,15,16]. Moreover, although many studies on woody plants have been carried out and inconsistent conclusions have been drawn, little attention has been paid to the interspecific variation of root traits of herbaceous plants [13,17,18]. Therefore, further research is necessary to verify the patterns of variation of root traits, which is necessary for us to understand the underground ecological process.

The difference of root traits represents the wide variations that species adapt to biotic and abiotic factors, which is closely related to the strategy of plant exploration and utilization of resources [19,20,21]. Root morphological traits, such as SRL and specific root area (SRA), are closely related to the capture efficiency of water and nutrients in soil [22]. In addition, root tissue density (RTD) is closely related to the defense of root [21,23]. Furthermore, some studies have shown that increasing root diameter (RD) can significantly improve the ability of roots to penetrate the soil [13,24]. Different correlations and combinations manifest in manner of root system to capture and store resources [17].

Whole plants can cope with possible environmental stress by changing the proportion of organ biomass and morphological traits [25,26,27]. Plants achieve different ecological adaptation by balancing the biomass distribution of different organs [26]. For instance, the high proportion of leaf biomass indicates high photosynthesis, but it is also accompanied by high water loss [28,29]. Plants, therefore, mitigate the threat of water loss from leaves by absorbing water to the greatest extent, which always requires higher efficiency of roots’ water uptake [30]. Previous studies often focused on the adaptation of plant biomass allocation patterns to the environment, and rarely integrated biomass allocation and root traits. In fact, the variation of plant morphological traits positively affects the manners in which they acquire resources and is ultimately manifested in biomass accumulation [31]. The investment of biomass obtained by different organs will become a powerful driving force for the morphological changes of various organs in the future [32]. Consequently, it is imperative to synthetically analyze the variation of morphological traits and biomass allocation characteristics to help us further understand the ecological adaptation of plants to the environment.

Annual ephemerals, an important plant group in the desert of Northern Xinjiang Uygur Autonomous Region, are the primary components of early spring vegetation in this region [33]. They are pioneer species in the restoration process of degraded desert areas, which play a considerable role in the stability of sand dunes and can reduce the frequency and intensity of dust storms [33,34,35]. They mostly grow in the loose sand (LS) soil of sand dunes, and some grow in the dense or gravelly sand (DGS) soil [36]. As a consequence, on the basis of insights into root traits and biomass allocation provided by available studies, this paper (i) determined whether the relationship between root traits in annual ephemeral species aligns with the assumption of root economics spectrum. Moreover, we hypothesize that (ii) species rooting DGS and species rooting LS manifest convergent adaptation, which will adopt different root strategies due to different soil pressures and tend to adopt different ecological adaptation strategies because of the trade-off between growth and survival. The research results can not only reveal the adaptation strategies of ephemeral plants to desert environments, but also provide a theoretical basis for the restoration and protection of desert vegetation. Finally, it will enrich the global root database and provide some data support for building a complete and more extensive root theory.

## 2. Results

### 2.1. Correlation among the Root Traits

The results of Pearson’s correlation analysis are shown in Figure 1. In terms of the root morphological traits mentioned in the literature that play a major function, SRA was positively correlated with SRL and negatively correlated with RTD (*p* < 0.05). The RD was negatively correlated with SRL and RTD (*p* < 0.05), but not with SRA (*p* > 0.05). The SRL was not correlated to RTD (*p* > 0.05). Maximum root depth (MRD) was significantly negatively correlated with RD and positively correlated with SRL and SRA (*p* < 0.05). The MRD was not significantly correlated to RTD (*p* > 0.05).

### 2.2. Comparison of Root Morphology Traits under Two Soil Conditions

In this study, the root morphology of annual ephemeral species rooting different soil conditions was compared by independent sample *t*-test. The results showed that the MRD of species rooting LS was significantly higher than that of species rooting DGS (Figure 2A). In addition, the SRL of the species rooting LS was also significantly higher than that of the species rooting DGS (Figure 2B). However, the RD of species rooting LS is significantly lower than that of species rooting DGS (Figure 2C). Finally, there was no significant difference in SRA and RTD between the two soil conditions (Figure 2D,E).

### 2.3. Correlation between Whole-plant Morphology Traits and Biomass Allocation

For the whole plant, we employed principal component analysis (PCA) to evaluate the correlation between traits, focusing on 10 traits including biomass fraction (Figure 3A). The first two axes together accounted for 51.6% of the variation. The PC1 axis and PC2 axis explained 32.5% and 19.1% of the variation, respectively (Figure 3A). The species were mainly distributed along the PC1 axis (Figure 3B). Those species that allocate biomass to aboveground light-trapping organs and have higher acquisition efficiency of underground resources are distributed on the right side of the axis, which have higher leaf mass fractions (LMF), MRD, and SRL. The species on the left side of the axis with higher CD and RD distributed more biomass in the stem and showed higher efficiency of nutrient transport and ability of conservation in growth (Figure 3A). In general, the PC1 axis lists the species with high resource retention capacity on the left and the species with high resource demand on the right (Figure 3B). There was only a significant difference between LS and DGS species on the whole-plant PC1 axis (Table 1).

## 3. Discussion

### 3.1. Correlation among Root Traits and Root Economics Spectrum

The correlation between plant traits (Figure 1) may be that natural selection makes some traits have a joint response to the environment, which is a trait-adaptive strategy for plants to adapt to the environment [37,38]. This adaptability is a combination of traits gradually formed after natural selection in the long-term evolution of plants, which indicates the ecological strategy of plants to adapt to the environment [39]. In this investigation, there were universal correlations among the five main root traits. Aligning with the hypothesis of RES, the significant positive correlation between SRA and SRL indicates that the longer the root length per unit weight (Figure 1), the larger the surface area [40]. Higher root length and surface area per unit biomass investment makes plants have high resource acquisition efficiency [22], but this is at the cost of reducing root tissue lifespan because tissues with low tissue density usually have shorter lifespans [12].

The assumption of the root economics spectrum is that RD and RTD belong to conservative economic strategies, so they should be positively correlated [10]. However, the present research and other experiments show that RD is significantly negatively correlated with RTD (Figure 1) [13,41,42], and there is no linear relationship [43]. Moreover, some studies believe that roots play a variety of physiological functions in complex soil environments, such as absorbing water and nutrients, interacting with microorganisms, and secreting root exudates [16,44]. Therefore, a one-dimensional economic spectrum of resource acquisition and conservation cannot be used to define the variation of root traits [12,16]. The variation of root traits among species may be multi-dimensional [4]. In our study, this is also proven by the inconsistent correlation between RD and SRL and SRA (Figure 1). To sum up, these experimental data do not support our hypothesis that root traits of annual ephemeral species coordinate along a one-dimensional RES and may even be multidimensional.

### 3.2. Differences in Root Strategies between Species Rooting LS and DGS

The variation of root distribution and root morphology reflects the adaptability of different species to soil water and nutrient availability [45]. Different root morphologies indicated different patterns of acquiring resources and defending environmental restrictions [46]. The species rooting LS have higher SRL (Figure 2B), which helps the root system to improve the absorption efficiency of water and nutrients in the soil [47,48]. Their long root system increases the possibility of meeting infiltrating water and nutrient patches [23,49]. Moreover, the species rooting DGS have lower MRD (Figure 2A). Their roots are mostly distributed in the shallow layer, mainly using short rain season and unpredictable rainfall [50].

In terms of defense, the species rooting DGS have higher RD (Figure 2C), which helps their roots to penetrate the soil with higher density [21]. This may be because they grow in the soil with high soil restriction. It is necessary to increase the root diameter by sacrificing SRL to ensure the anchoring and survival of plants in the soil [24]. We predict that DGS species face severe soil constraints, so their roots have high defense capabilities that help them alleviate the restriction of soil on root development [51,52]. That is, species rooting DGS have higher RD or RTD (Figure 2C). Because thick roots have a large cross-sectional area of the stele, this kind of root can be protected from mechanical damage, herbivores, and drought stress [13,52,53]. Our data partially support our hypothesis that species rooting DGS have high RD because they live with high soil disturbance (Figure 2C). However, our experimental data do not support our hypothesis that species rooting DGS have higher RTD (Figure 2D), because plants growing under two soil conditions have similar RTD. This may be, as found in the trees, because the way herbaceous plants construct roots is not limited by RTD [12].

Plants have a common trade-off between growth and survival [13]. Species rooting DGS tend to adopt a root strategy with high soil penetration and low resource acquisition efficiency, which effectively ensures their survival in more challenging soil conditions. In addition, the higher MRD and SRL of species rooting LS enable them to efficiently explore and exploit resources in soil, which is an effective resource utilization strategy.

### 3.3. Differences in Whole-plant Strategies between Species Rooting LS and DGS

Terrestrial vascular plants have typical root, stem, and leaf structures [48]. The leaves are responsible for the fixation of C, the stems provide mechanical support and hydraulic channels, and the roots absorb water and nutrients in the soil and play an anchoring role [54]. As a consequence, in order to maintain necessary physiological activities and achieve normal growth, plants must balance the biomass distribution of leaves, branches, stems, and roots [55,56]. Additionally, plants achieve the balance of resource acquisition and allocation through coordinated changes in biomass allocation and morphological traits [57,58].

In the present investigation, how species are separated along the PC1 axis indicates the changes in ecological strategies of different species (Figure 3A). For the 50 annual ephemeral species we studied, the species on the left side of the whole-plant PC1 axis show a resource-conservative strategy on the whole plant. These plants have a higher biomass fraction to nutrient transport organs (stems) ratio, and they have thicker roots and higher root collar diameters (Figure 3). The lower proportion of root biomass allocation indicates that these plants have weak demand and competitiveness for nutrients [48,59]. The limited carbon obtained by them is mainly used for the growth and reproduction of aboveground parts, rather than the acquisition of underground resources [60]. DGS species tend to elect this whole-plant strategy (Figure 3).

The species at the other end of the axis have a higher biomass fraction to resource acquisition organ ratio, which indicates that they have higher requirements for water and nutrients (Figure 3). Their roots can usually forage in deeper soil layers and satisfy their high demand for water and nutrients by improving SRL and SRA [18,30]. This helps their growth and survival because snow melt serves as a water supply in winter and spring in deep sand soil, and the lower temperature means the water can be stored for a longer time [50]. Moreover, the relatively high LMF of these plants makes them have high water consumption [31] because the fixation of C is unavoidable when accompanied by water loss [28,29]. Their higher RMF and higher water absorbance efficiency appear to alleviate the higher water consumption of leaves (Figure 3) [30]. The species rooting LS tend to adopt this whole-plant strategy.

## 4. Materials and Methods

### 4.1. Geography of the Study Area

The Junggar Basin (34°09′–49°08′ N, 73°25′–96°24′ E) is located in the north of Xinjiang Uygur Autonomous Region, China. It is situated between the Altai and Tianshan Mountains, forming an irregular triangular terrain. It is 1100 km from east to west and as much as 800 km wide from north to south. The Gurbantonggut Desert is located in the hinterland of Junggar Basin, covering an area of 48.8 thousand km^2^. The area is far from the sea, with annual evaporation of 1400–1700 mm and annual average rainfall of no more than 200 mm. There is relatively stable snow in winter and the annual average temperature is −4 to 9 °C, which is a typical cold desert [33]. The vegetation is mainly composed of shrubs (e.g., *Haloxylon ammodendron*), perennial herbs (e.g., *Astragalus flexus*, *Eremurus inderiensis* et al.), and annual ephemerals [34]. The precipitation distribution in spring and summer is higher than that in autumn and winter, which provides favorable growth conditions for a substantial number of ephemerals. In the season with the most vigorous growth of ephemerals (late April/early May), the coverage rate can reach 40% [34]. Consequently, annual ephemeral species are the dominant plant group in the Gurbantunggut Desert. They mostly grow in the loose sand soil of sand dunes, and some grow in the gravelly or highly compacted sand soil [36]. On this basis, we divided the collected 50 ephemerals species into those rooted in dense or gravelly sand (DGS) or loose sand (LS) groups according to different soil conditions. The root system of the DGS group has to deal with the constraint of gravel or high-density sand soil during development while the LS group could not handle an abominable soil environment.

### 4.2. Field Investigation and Sample Collection

Samples of ephemerals were harvested from April to May 2022. Microhabitat information of the collection site, such as habitat type, longitude and latitude, altitude, and slope, were recorded. In the DGS and LS soil environments, we collected 16 species and 34 species, respectively. It is worth noting that two species (viz., *Lappula semiglabra* and *Plantago minuta*) exist simultaneously in different environments, and we regard them as different species for calculation. The 16 species rooting DGS and 34 species rooting LS were collected from 4 quadrats, respectively, with at least 3 plants distributed in each quadrat (Table 2). At each sampling site, the area with similar terrain and relatively uniform plant distribution was chosen to set a 10 m × 10 m quadrat to investigate the growth indicators (e.g., plant height, root collar diameter) of each species in the quadrat to calculate the standard plant. Considering that the roots of annual ephemeral species are mostly distributed in the shallow soil layer, our excavation depth was 30 cm below the soil surface.

In each plot, ten individuals of each species were selected with aboveground trait values similar to the standard plant, and all the roots of each plant were excavated in situ with large shovels, small shovels, a 30 cm steel ruler, a brush, and other tools. The specific method of digging is to dig a pit that is 10 cm long, 5 cm wide, and 40 cm deep with a large shovel 3–4 cm from the plant. Subsequently, the maximum rooting depth (MRD) was measured by steel tape. After the measurements, the soil surrounding the roots was carefully cleaned into deep pits until the roots were thoroughly exposed. Then, the whole plant was cut at the base of the stem (root collar) with scissors, and the above-ground portion of the plant was stored in an envelope. The root systems were stored in a plastic bag in an ice box, and the identifier was recorded on the envelope and the plastic bag. The number consists of a serial number of collected samples, the species name, and sampling repetition number (i.e., 1 to 10).

### 4.3. Trait Measurements and Calculations

The root samples were transported to the laboratory and the sand soil adhering to the root surface was washed with deionized water to examine the relevant indexes. The washed roots were scanned with a scanner (Epson Perfection V850 Pro Photo Scanner; Epson, Los Alamitos, CA USA) to obtain images of complete roots, which were stored in the computer by identifiers analogous to the root samples stored in plastic bags. The white or blue background plate was selected according to the color of the root surface. Before scanning, the surface of the background plate was inspected to ensure that it was smooth and free of impurities. The white background plate was selected for those with a darker root color and the blue background plate for those with a lighter root color to improve the contrast. The purpose of this was to improve the contrast between the root system and the background plate and facilitate subsequent analysis. The root image was analyzed by root analysis system software Win RHIZO Pro 2013 (Regent Instruments, Inc., Canada; Available online: www.regentinstruments.com (accessed on 24 August 2022)) to compute the data of total root length (RL), root surface area (RA), root diameter (RD), and root volume (RV). After scanning, the roots were dried in an 80 °C oven and weighed to calculate the dry weight. The specific root length (SRL) is calculated as the ratio of total root length to biomass, the specific root area (SRA) is surface area to biomass, and the root tissue density (RTD) is biomass to root volume (Table 3).

The aboveground part of the recovered plant was divided into stem and leaf parts, which were separately packed in small envelopes and dried at 65 °C for 48 h until reaching a constant weight, after which the mass of each part was weighed. Finally, the root mass fraction (RMF), leaf mass fraction (LMF), and stem mass fraction (SMF) were calculated (Table 3).

### 4.4. Data Analysis

Limited by the sampling standard of species distribution, we only set four quadrats for two soil conditions, and each quadrat had at least three species of plants distributed. For each species, we collected 10 samples and calculated their average traits for later analysis. The values of each trait were transformed by log10 to assume normality and homogeneity of variance. Pearson’s correlation analysis was used to evaluate the relationship between whole-plant traits. Based on the results of the correlation analysis, we conducted further linear regression analysis on the related root traits. Furthermore, Principal component analysis was used to evaluate the relationship between root traits and whole-plant traits of each species (Table 2). The difference between species rooting LS and species rooting DGS on the two axes was evaluated by an independent sample *t*-test. All statistical analyses were performed in SPSS 26.0. We used origin 2021 and ggplot2 (R 4.0.3) to perform data visualization.

## Figures and Tables

**Figure 1 plants-11-02495-f001:**
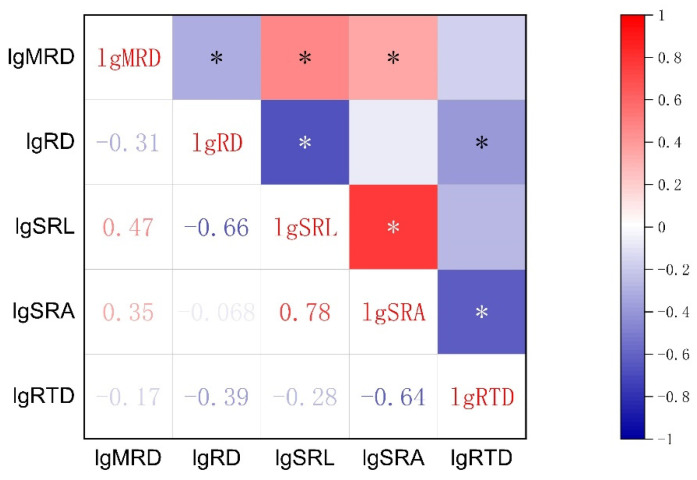
Correlation among the root traits of 50 ephemeral plants growing in two soil environments (loose sand or dense, gravelly sand) in the cold desert of the Chinese Junggar Basin in 2022. The color gradually changes from blue to red indicating that the inter-trait correlation changes from negative to positive correlation, with the darker color indicating the stronger the correlation. The values of each root trait were transformed by log10 to assume normality and homogeneity of variance. The meanings of MRD, RD, SRL, SRA, and RTD are maximum root depth, root diameter, specific root length, specific root area, and root tissue density, respectively. * indicates *p* < 0.05.

**Figure 2 plants-11-02495-f002:**
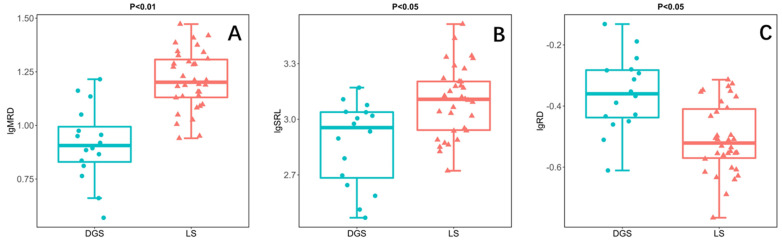
Results of independent sample *t*-test illustrating the differences between species rooting in loose sand (LS; n = 34) or rooting in dense or gravelly sand (DGS; n = 16) in the in the cold desert of the Chinese Junggar Basin in 2022. (**A**) difference in maximum rooting depth (MRD), (**B**) difference in specific root length (SRL), (**C**) difference in root diameter (RD), (**D**) difference in root tissue density (RTD), (**E**) difference in specific root area (SRA). *p* < 0.05 indicates significant difference between species rooting LS and species rooting DGS. The values for each species represent 10 plants.

**Figure 3 plants-11-02495-f003:**
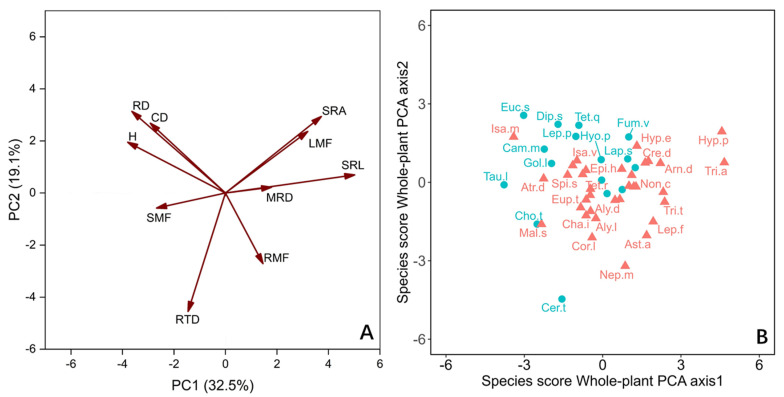
Principal component analysis (PCA) with the mean values (n = 10) of the whole-plant traits, of 50 annual ephemeral species growing in either loose sand or dense, gravelly sand in the cold desert of the Chinese Junggar Basin in 2022. (**A**) PCA with the mean of 10 values of the whole-plant traits. (**B**) Score of species along the first two axes of whole plant. Species names are given as abbreviations (see Table 2). The red triangles and ellipses represent species rooting in loose sand (LS). The blue dots and ellipses represent species rooting in gravelly or dense sand (DGS). The meanings of H, CD, MRD, RD, SRL, SRA, RTD, LMF, SMF and RMF are plant height, root collar diameter, maximum root depth, root diameter, specific root length, specific root area, root tissue density, leaf mass fraction, stem mass fraction, and root mass fraction, respectively.

**Table 1 plants-11-02495-t001:** Differences in principal component (PC) 1 and PC2 scores between ephemeral species rooting in loose sand (LS; n = 34) or gravelly or dense sand (DGS; n = 16) and (mean ± SE) in the cold desert of the Chinese Junggar Basin in 2022.

PCA	Axis	Rooting Soil Conditions		Significance Test	
		Rooting LS	Rooting DGS	*t*	*P*
Whole-plant	PC1	0.43 ± 0.40 ^a^	−0.92 ± 0.39 ^b^	2.61	0.01
	PC2	−0.23 ± 0.20 ^a^	0.50 ± 0.43 ^a^	−1.79	0.08

Different lowercase letters in the same row indicate significant differences among species (*p* < 0.05).

**Table 2 plants-11-02495-t002:** List of 50 annual ephemeral species in the cold desert of the Chinese Junggar Basin in 2022. LS represents the species rooting in loose sand, and DGS represents the species rooting in dense or gravelly sand. For each species, we calculate the average value of each trait. The meanings of H, CD, MRD, RD, SRL, SRA, RTD, LMF, SMF, RMF are plant height, root collar diameter, maximum root depth, root diameter, specific root length, specific root area, root tissue density, leaf masss fraction, stem mass fraction, and root mass fraction, respectively.

Species	Family	Code	Group	H (cm)	CD (mm)	MRD (cm)	RD (mm)	SRL (cm g^−1^)	SRA (cm^2^ g^−1^)	RTD (g cm^−3^)	LMF (g g^−1^)	SMF (g g^−1^)	RMF (g g^−1^)
** *Descurainia sophia* **	Brassicaceae	**Des.s**	LS	19.47	1.29	19.37	0.29	1239.28	111.32	1.36	0.21	0.46	0.24
** *Plantago minuta* **	Plantaginaceae	**Pla.m**	DGS	3.24	1.232	13.65	0.37	1091.74	113.78	1.07	0.58	0.13	0.12
** *Plantago minuta* **	Plantaginaceae	**Pla.m**	LS	6.86	2.407	12.52	0.28	1337.01	114.67	1.44	0.46	0.05	0.26
** *Chamaesphacos ilicifolius* **	Lamiaceae	**Cha.i**	LS	10.38	1.049	13.78	0.38	1161.03	139.46	3.01	0.27	0.32	0.08
** *Nepeta micrantha* **	Lamiaceae	**Nep.m**	LS	5.33	0.552	8.93	0.20	1279.70	80.53	2.49	0.29	0.19	0.28
** *Euphorbia turczaninowii* **	Euphorbiaceae	**Eup.t**	LS	5.38	0.913	19.84	0.43	868.70	114.24	0.85	0.21	0.51	0.19
** *Trigonella arcuata* **	Fabaceae	**Tri.a**	LS	2.9	0.775	25.62	0.23	2737.76	199.92	0.88	0.59	0.07	0.17
** *Astragalus arpilobus* **	Fabaceae	**Ast.a**	LS	3.67	0.737	22.13	0.29	1078.93	100.08	1.59	0.40	0.15	0.35
** *Tribulus terrestris* **	Zygophyllaceae	**Tri.t**	LS	0.99	0.755	16.24	0.32	1874.33	186.86	0.70	0.29	0.30	0.31
** *Centaurea pulchella* **	Asteraceae	**Cen.p**	LS	9.46	1.063	21.94	0.31	1507.41	142.75	0.95	0.53	0.22	0.12
** *Crepis desertorum* **	Asteraceae	**Cre.d**	LS	8.59	0.724	10.62	0.31	2198.19	202.17	0.78	0.26	0.23	0.06
** *Amberboa turanica* **	Asteraceae	**Amb.t**	LS	9.28	1.425	20.44	0.39	1100.29	127.31	0.87	0.18	0.41	0.08
** *Epilasia hemilasia* **	Asteraceae	**Epi.h**	LS	9.03	1.1	19.24	0.49	865.36	132.71	0.78	0.26	0.37	0.13
** *Koelpinia linearis* **	Asteraceae	**Koe.l**	LS	9.18	0.861	15.39	0.31	1269.16	122.90	1.24	0.27	0.25	0.16
** *Lactuca undulate* **	Asteraceae	**Lac.u**	LS	11.21	1.623	11.22	0.45	728.92	93.23	1.14	0.37	0.25	0.08
** *Senecio subdentatus* **	Asteraceae	**Sen.s**	LS	12.12	1.236	21.2	0.32	1516.69	147.62	0.97	0.32	0.25	0.11
** *Atriplex dimorphostegia* **	Amaranthaceae	**Atr.d**	LS	17.06	1.232	13.64	0.41	744.77	92.18	1.15	0.21	0.50	0.05
** *Corispermum lehmannianum* **	Amaranthaceae	**Cor.l**	LS	9.01	0.967	18.74	0.24	1042.88	78.61	2.66	0.32	0.38	0.12
** *Ceratocephala testiculata* **	Ranunculaceae	**Cer.t**	DGS	5.13	0.427	3.71	0.25	326.38	24.44	7.53	0.07	0.15	0.14
** *Hyoscyamus pusillus* **	Solanaceae	**Hyo.p**	DGS	7.72	1.282	7.67	0.44	1096.56	151.10	0.97	0.31	0.18	0.08
** *Tetracme quadricornis* **	Brassicaceae	**Tet.q**	DGS	11.34	2.562	9.44	0.52	945.80	239.94	1.38	0.29	0.29	0.09
** *Lachnoloma lehmannii* **	Brassicaceae	**Lac.l**	DGS	6.76	1.113	7.34	0.37	1012.26	112.73	1.06	0.39	0.27	0.11
** *Chorispora sibirica* **	Brassicaceae	**Cho.s**	DGS	5.69	0.842	11.24	0.36	1193.59	125.86	1.47	0.44	0.23	0.12
** *Lepidium apetalum* **	Brassicaceae	**Lep.a**	DGS	8.32	1.154	14.52	0.43	859.89	117.50	1.69	0.46	0.23	0.22
** *Lepidium perfoliatum* **	Brassicaceae	**Lep.p**	DGS	8.51	1.238	4.58	0.65	615.42	112.18	0.81	0.45	0.13	0.06
** *Chorispora tenella* **	Brassicaceae	**Cho.t**	DGS	12.03	1.177	6.48	0.49	441.59	66.02	2.96	0.15	0.19	0.07
** *Camelina microcarpa* **	Brassicaceae	**Cam.m**	DGS	19.13	1.628	8.92	0.52	497.92	78.90	1.09	0.43	0.30	0.07
** *Goldbachia laevigata* **	Brassicaceae	**Gol.l**	DGS	24.9	1.49	7.83	0.41	788.01	95.33	1.11	0.28	0.23	0.10
** *Euclidium syriacum* **	Brassicaceae	**Euc.s**	DGS	21.46	1.807	8.29	0.74	386.88	86.44	0.66	0.33	0.15	0.04
** *Diptychocarpus strictus* **	Brassicaceae	**Dip.s**	DGS	25.23	1.718	9.04	0.51	1045.39	169.57	0.67	0.24	0.26	0.07
** *Tauscheria lasiocarpa* **	Brassicaceae	**Tau.l**	DGS	23.2	1.595	5.81	0.57	294.54	52.59	1.76	0.18	0.31	0.12
** *Malcolmia scorpioides* **	Brassicaceae	**Mal.s**	LS	21.45	1.913	19.32	0.28	708.90	61.81	3.07	0.30	0.42	0.13
** *Alyssum linifolium* **	Brassicaceae	**Aly.l**	LS	16.92	1.354	16.24	0.23	1415.04	100.22	2.02	0.29	0.34	0.20
** *Alyssum dasycarpum* **	Brassicaceae	**Aly.d**	LS	14.78	1.47	15.56	0.24	1283.34	93.89	2.46	0.33	0.28	0.09
** *Leptaleum filifolium* **	Brassicaceae	**Lep.f**	LS	7.17	0.55	8.71	0.17	2164.55	114.37	2.18	0.38	0.14	0.11
** *Isatis violascens* **	Brassicaceae	**Isa.v**	LS	18.43	1.253	16.95	0.45	773.55	105.26	0.89	0.43	0.34	0.15
** *Cithareloma vernum* **	Brassicaceae	**Cit.v**	LS	5.4	0.819	14.35	0.31	1316.82	117.72	1.31	0.35	0.31	0.13
** *Isatis minima* **	Brassicaceae	**Isa.m**	LS	26.53	4.036	23.71	0.47	527.79	71.85	1.28	0.33	0.33	0.20
** *Spirorhynchus sabulosus* **	Brassicaceae	**Spi.s**	LS	16.81	1.551	24.31	0.46	775.16	95.97	1.00	0.29	0.38	0.19
** *Tetracme recurvata* **	Brassicaceae	**Tet.r**	LS	10.33	1.213	10.13	0.37	670.63	77.79	1.67	0.46	0.20	0.14
** *Silene olgiana* **	Caryophyllaceae	**Sil.o**	LS	8.51	1.027	14.32	0.27	1476.95	123.35	1.33	0.46	0.19	0.14
** *Fumaria vaillantii* **	Papaveraceae	**Fum.v**	DGS	10.71	1.233	6.85	0.35	1482.44	157.62	0.76	0.59	0.22	0.04
** *Hypecoum erectum* **	Papaveraceae	**Hyp.e**	LS	19.15	1.442	29.69	0.28	1601.33	137.29	1.06	0.57	0.19	0.09
** *Hypecoum parviflorum* **	Papaveraceae	**Hyp.p**	LS	7.39	0.68	15.46	0.28	3262.09	286.09	0.55	0.48	0.16	0.12
** *Lappula semiglabra* **	Boraginaceae	**Lap.s**	DGS	10.52	1.676	16.42	0.31	1281.78	119.61	1.36	0.59	0.14	0.09
** *Heliotropium acutiflorum* **	Boraginaceae	**Hel.a**	LS	6.71	1.039	12.09	0.44	893.38	122.95	0.85	0.26	0.37	0.19
** *Nonea caspica* **	Boraginaceae	**Non.c**	LS	8.98	0.873	26.25	0.25	1944.64	148.63	1.12	0.38	0.19	0.21
** *Arnebia decumbens* **	Boraginaceae	**Arn.d**	LS	6.39	1.038	15.22	0.28	2131.62	182.87	0.82	0.41	0.23	0.11
** *Lappula semiglabra* **	Boraginaceae	**Lap.s**	LS	11.24	1.201	12.3	0.27	1659.34	131.37	1.25	0.47	0.22	0.12
** *Lappula lasiocarpa* **	Boraginaceae	**Lap.l**	LS	11.58	1.126	13.47	0.25	1594.84	124.31	1.61	0.48	0.18	0.13

**Table 3 plants-11-02495-t003:** Description of morphological traits measured on ephemeral species rooting in loose sand or gravelly or dense sand in the cold desert of the Chinese Junggar Basin in 2022.

Abbrev.	Trait	Unit	Implication
CD	Root collar diameter	mm	Reflecting the transportation efficiency of root nutrients and water to the aboveground part of plant
H	Plant height	cm	Plant height is related to plant longevity and the potential to compete for sunlight
MRD	Maximum root depth	cm	Reflecting the explored potential of root to soil layer
RD	Root diameter	mm	Reflecting the penetration of root system to soil
SRL	Specific root length	cm g^−1^	The root length per biomass investment is closely related to the efficiency of plants in capturing water and nutrients
SRA	Specific root area	cm^2^ g^−1^	The root surface area per biomass investment is closely related to the efficiency of plants in capturing water and nutrients
RTD	Root tissue density	g cm^−3^	The root biomass investment per volume can reflect the tensile strength and defensive strength of roots.
LMF	Leaf mass fraction	g g^−1^	The biomass assigned to leaves by plants for photosynthesis.
SMF	Stem mass fraction	g g^−1^	The biomass allocated to stem by plants for Supporting leaves and transporting water and nutrients between roots and leaves.
RMF	Root mass fraction	g g^−1^	The biomass investment of plants in underground foraging.

Implications of traits are based on [11,18,21,22,48,61,62].

## Data Availability

The processed data required to reproduce these findings cannot be shared at this time as the data also forms part of an ongoing study.

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
