# Peer review of "Root Morphology and Biomass Allocation of 50 Annual Ephemeral Species in Relation to Two Soil Condition"

_plants, 2022, doi:10.3390/plants11192495_

Round 1
Reviewer 1 Report
The present work is very interesting and informative. The authors presented their works very well and the presentation of their research findings accurately. But I have given some minor corrections-
1. The title can not reflects the findings.
2. The abstract should be write lucrative way and precisely and shortly.
3. The quality of the figures are not good. Better to change new figures.
Reviewer 2 Report
The present study highlighted the biomass allocation and root dynamics of common ephemerals in cold dessert. From the perspective of ecosystem dynamics, the study is valuable, informative, and well written. Although the data is limited for a full-length paper but given the importance of the topic and good statistical analysis work, I suggest minor revisions
1. Line 9-10: What species?
2. Line 88-93: This part belongs to material & methods and discussion section.
3. Section 4.1 may be written from the perspectives of ephemerals. The local vegetations may be described in few sentences.
4. The future implications of this work should also reflect in abstract, introduction and conclusion part.
Reviewer 3 Report
Comments to Authors: MDPI Plants-1908098, Root morphology and biomass allocation of 50 annual ephemerals in cold desert
The manuscript is fairly well-written and organized and the English grammar only needs minor improvement. Thank you! There are no major concerns, but line-by-line comments are provided below for improvement:
General:
Somewhere, and more than once, state in the text that there were 16 DGS and 34 LS species with no species common to both soil environments. Throughout the manuscript, I was under the impression that the same 50 species were found in both soil environments. It was not until I reviewed supplemental table 1 that I realized the situation despite what it says at Lines 21-22, which says you divided the species by soil environment. Since it could be argued that you did not divide them, but rather that was their natural environment where you found them, that statement should be revised.
Lines 21-22 and elsewhere: “. . . species into those rooted in gravelly and dense sand (DGS) or loose sand (LS) groups . . .” The fully spelled term is not parenthesized; the abbreviation is to follow the spelled out term and be in parentheses. Also, since the abbreviation is DGS, should the original term not be “dense and gravelly sand”? Finally, at Lines 21-22 and many other places “and” is used between “gravelly” and “dense” while in other places it is “gravelly or dense”. The meanings are different. If the environment was both dense AND gravelly, “and” is correct; if you had some plots with gravel and others with dense sand, “or” is appropriate. This must be clarified at all points. Consistency in terminology is important throughout the manuscript.
Line 34: “. . . capacity that the root system . . .”
Line 39: “Therefore, the root system . . . plays and extremely . . .”
Line 43: “. . . spectrum (RES) indicates . . .” Abbreviations are to be defined on first use in the Abstract, main text, AND every table and figure where they are used.
Line 44: “. . . (SRL) means resource . . .”
Line 49: “. . . does not agree with . . .” “Concur” also would be an appropriate word.
Line 50: “. . . of the root . . .”
Line 55: “. . . root traits, which . . .”
Line 59: specific root length was defined in the main text at line 44. Delete the spelled out term here and use the abbreviation consistently throughout the remainder of the manuscript. Remember that it must be defined in every table or figure in which the abbreviation is used.
Lines 59-61: This is an incomplete sentence. It appears as though it should say, “Root morphological traits, such as SRL and specific root area (SRA) are closely related to the capture efficiency of water and nutrients in soil [22].”
Line 61: “. . . closely related to the . . .”
Line 64: “. . . correlations and combinations manifest in the manner of root systems to . . .”
Line 66: “Whole plants can cope . . .”
Line70: “Plants, therefore, mitigate . . .”
Line 72: “. . . efficiency of root water uptake [30].”
Line 80: “. . . ephemerals, an important plant group . . .”
Line 83: “ephemerals”
Line 84: “. . . 14.1%, respectively.”
Line 87: This is spelled differently at line 254. Check all usages and be consistent in spelling.
Lines 88-90: See comments about Lines 21-22. Be consistent.
Lines 91-93: “The root system of the DGS group . . . during development while the LS group could not handle abominable soil environments.”
Line 103: . . . analysis are shown in Figure 1.”
Line 104: “literature”.
Line 106: “The RD was . . .” Do not begin a sentence with an abbreviation, acronym, or numeral. If the sentence cannot be revised to avoid this, usually like adding an article (a, an, the) to begin the sentence, spell out the full term to begin the sentence.
Line107: “The SRL was not . . .” Use of abbreviation to begin a sentence and use of present tense to describe results. Generally, methods and results are reported in the past tense. This will be the final mention of these issues: check throughout the manuscript for abbreviations to begin sentences and use of present vs. past tense.
Figure 1 caption: “Correlation among the root traits of 50 ephemeral plants growing in two soil environments (loose sand or dense, gravelly sand) in the cold desert of the Chinese Junggar Basin in 2022. The color gradually changes from blue to red indicating that the inter-trait correlation changes from negative to positive correlation, with the darker color indicating the stronger the correlation. The values of each root trait were transformed by log10 to assume normality and homogeneity of variance. The meanings of MRD, RD, SRL, SRA, and RTD are maximum root depth, root diameter, specific root length, specific root area, and root tissue density, respectively. * indicates P<0.05.” Remember, all abbreviations must be defined on first use in the Abstract, main text, AND each figure or table in which it is used. Tables and figures must stand alone from each other and the text because other scientists and educators may copy and use them in their work. Also, in each table title or figure caption, state the location of the study. In the example of that I showed above, I also indicated the described the general environmental conditions. Finally, including the number of plant species provides a basis for the replication in the correlation analysis. If the means of the 50 species was not the basis, but, rather, the measurements from individual plants, show “(n = the total number of plants collected and measured)” immediately after “Correlation”.
Lines 119-122: “. . . MRD of LS species was significantly higher than LS species.” In addition to the suggested revision of the sentence for conciseness (“LS species” compared to “species rooting LS”), which should be revised throughout the manuscript, note that the sentence makes a comparison of LS species. This must be clarified. Finally, it is not necessary to report the t-statistic for comparisons of two treatments. The p-value is adequate. In this case the p-value also is provided on the figure, which is good, but do not report information in the text that is already reported in a table or figure. These comparisons are reported as significant at P<0.01 in the text, but P<0.05 in the figure. Be consistent, but since you should report data and statistical results in one place, it is more important to simply be correct.
Figure 2 caption: “Results of independent sample t-test illustrating the differences between species rooting in loose sand (LS; n = 34) or rooting in gravely or dense sand (DGS; n = 16) in the in the cold desert of the Chinese Junggar Basin in 2022. A) difference in maximum rooting depth (MRD), B) difference in specific root length (SRL), C) difference in root diameter (RD), D) difference in root tissue density (RTD), E) difference in specific root area (SRA). P<0.05 indicates significant difference between LS species and DGS species. The values for each species represent 10 plants.” There will be a question in the methods about how many plots of each soil environment were sampled. If there was more than one plot, the last sentence should read, “The values for each species represent 10 plants harvested from x plots of each soil environment.” It is very important to be clear about replication. If there were a major concern about this manuscript, it would be that it is not clear how many plants were actually sampled. Manuscripts are to contain enough information to satisfy readers that sufficient data were collected to support discussion and conclusions/implications, as well as to repeat the study if they desire.
Figure 3: First move to follow the first time called out, which is at current line 147. Caption: “Principal component analysis (PCA) with the mean values (n = 10) of the whole-plant traits, of 50 annual ephemerals growing in either loose sand or dense, gravelly sand in the cold desert of the Chinese Junggar Basin in 2022. A) PCA with the mean 5 of 14 values of the whole-plant traits. B) Score of species along the first two axes of whole-plant. Species names are given as abbreviations (see Table S1). The red triangles and ellipses represent species rooting in loose sand (LS). The blue dots and ellipses represent species rooting in gravely or dense sand (DGS). The meanings of H, CD, MRD, RD, SRL, SRA, RTD, LMF, SMF and RMF are height, root collar diameter, maximum root depth, root diameter, specific root length, specific root area, root tissue density, leaf mass fraction, stem mass fraction, and, root mass fraction, respectively.” Remember that defining abbreviations in figure captions and table titles that are used in the table or figure is necessary, but in the case of abbreviating the names of 50 species, that is probably not reasonable. So, perhaps referring to Table S1 is okay in this case, although it normally would not be. Nevertheless, readers should not have to search a supplemental table or figure for such critical information. Hence, a suggestion is made to include Table S1 as Table 3 in the main text and cite it in this caption and at the end of the first sentence on line 312. Note also that no ellipses are visible in the figure and that the x- and y-axis titles should be similarly formatted between panes A and B.
Line 146: “. . . employed principal component (PC) analysis . . .” The abbreviation of the term must be defined because the abbreviation is used at lines 148-149.
Lines 150-151: “. . .biomass to aboveground light trapping organs . . .”
Line 152: “. . . higher leaf mass fraction (LMF) . . .” All abbreviations must be defined on first use.
Lines 152-158: “The species on the left side of the axis with higher CD and RD, distributed more biomass in the stem and showed higher efficiency of nutrient transport and ability of conservation in growth (Fig. 3A). In general, the PC1 axis lists the species with high resource retention capacity on the 155 left and the species with high resource demand on the right (Fig. 3B). There was only significant difference between LS and DGS species on the whole-plant PC1 axis (Table 1).”
Table 1 title: “Differences in principal component (PC)1 and PC2 scores between ephemeral species rooting in loose sand (LS; n = 34) or gravely or dense sand (DGS; n = 16) and (mean ± SE) in the cold desert of the Chinese Junggar Basin in 2022.” Note gravelly or dense (also spelling of gravely compared to elsewhere) vs gravelly and dense. Also include the correct level of replication supporting each mean.
Line 166: “The correlation between plant traits (Fig. 1) may . . .” Use table or figure callouts whenever even potentially referring to your results. Use callouts early and often. You might also cite Figure 1 at the end of the sentence concluding on Line 171.
Line 173: “. . . area per unit biomass . . .”
Line 195: “. . . SRL (Fig. 2A), which . . .”
Line 205: Insert a figure callout after “RD or RTD”.
Line 229: “. . .investigation, how species are separated . . .”
Line 230: “. . . different species (Fig. 3A).”
Line 240: “. . .and nutrients (Fig 3).”
Line 248: “[30]. The LS species . . .”
Lines 252-253: Either, “. . . between the Altai and Tianshan Mountains, . . .” or “. . . between the Altai and Tianshan Mountain ranges, . . .”
Lines 253-254: “It is 1100 km from east to west and as much as 800 km wide from north to south.”
Line 255: “km2” Superscript the 2.
Lines 261-262: “. . . season with the most vigorous growth (late April/early May) with . . .”
Line 268: “recorded”. State how many sampling sites there were for each soil environment. You could either insert a sentence Stating how many, even if only one for each site or include the numbers parenthetically after “At each sampling site (x for LS and y for DGS) . . .” If there was only one: “At each sampling site (one each for LS and DGS).” This is critical information for interpreting the results and repeating the study.
Line 269: “. . . was chosen to set a 10mx10m quadrat to investigate the . . .”
Line 271: “. . . plant. Considering roots . . .”
Line 274: “. . . selected with aboveground trait . . .”
Line 277-278: “. . . a large shovel 3-4 cm from the plant.”
Line 288: “. . . root were transported to the laboratory and the sand soil adhering to . . .”
Line 290: “. . . scanner (Epson Perfection V850 Pro Photo Scanner; Epson, Los Alamitos, CA USA) to . . .”
Line 290: “. . . roots, which were stored in the computer . . .”
Lines 293-296: “Before scanning, the surface of the background plate was inspected to assure it was smooth and free of impurities. The white background plate was selected . . . and the blue background. . . color to improve the contrast . . .”
Line 298: “. . . 2013 (Regent Instruments, Inc., Canada; www.regentinstruments.com) to compute . . .” Providing the website when no city location is available lets the reader find more information, if desired.
Lines 300-301: “. . . roots were dried in . . . and weighed to calculate . . .”
Lines 304-309: “The aboveground part of the recovered plant was divided into stem and leaf parts, which were separately packed in small envelopes and dried at 65 ℃ for 48 h to constant weight, after which the mass of each part was weighed. Finally, root mass fraction (RMF), leaf mass fraction (LMF) and stem mass fraction (SMF) were calculated (Table 2).”
Line 312: Please state the level of replication and describe the statistical model. How many sites were evaluated for each soil environment? It is likely that if more than one site for each was evaluated, a completely randomized design would have represented the type of replication. Currently, 10 plants of each species were sampled at each LS and DGS location, but there were no species found at both environments. This must be clarified.
Table 2: Title: “Description of morphological traits measured on ephemeral species rooting in loose sand or gravely or dense sand in the cold desert of the Chinese Junggar Basin in 2022.” Not that LMF is defined as root mass fraction in the trait column, but in the Implication column “leaves” are mentioned.
It appears as though all references are properly formatted. Thank you for that!
The information collected during this study has potential to be of great value for the scientific community and the presentation in the manuscript has been fairly well done. Please follow the recommendations of the reviewers that are supported by the editors to revise this manuscript for publication.
